# Where to invest in neonatal survival programs in Nepal? A modelling study using Lives Saved Tool through scaling key interventions

Geha Nath Khanal[1,2]*, Nisha Giri[3], Deepak Jha[4], Dipak Raj Chaulagain[5]

1  Nepal Public Health Association, Lalitpur, Nepal, 2  School of Nursing, Midwifery, Allied and Public Health, Canterbury Christ Church University Canterbury, Canterbury, United Kingdom, 3  Nursing Association of Nepal, Kathmandu, Nepal, 4  Department of Health Services, Ministry of Health and Population, Family Welfare Division, Kathmandu, Nepal, 5  Institute for Implementation Science and Health, Kathmandu, Nepal

* khanalg@outlook.com

## Abstract

### Background

Neonatal survival remains a major public health priority in many low and middle-income countries including Nepal, where the neonatal mortality rate (NMR) is 21 per 1,000 live births. To improve the neonatal health outcomes, the World Health Organization (WHO) recommends scaling up coverage of quality antenatal care (ANC), skilled care at birth, immediate essential newborn care and resuscitation, postnatal care for mother and newborn, and specialized care of small and sick newborns. In this context, we aimed to estimate the potential impact of achieving the coverage targets for 35 interventions, identified as monitoring indicators under Nepal's Every Newborn Action Plan (NENAP), on neonatal survival in Nepal by 2035.

### Methods

We used the Lives Saved Tool (LiST) to estimate the number of neonatal deaths that could be prevented by 2035 if target intervention coverage level is achieved, starting from 2025 baseline values. Baseline data were drawn from demographic and health (DHS) survey and multiple indicator cluster survey (MICS). Where survey data were unavailable, proxy estimates were derived from available literatures to establish the baseline levels.

### Results

By 2035, achieving the target coverage level could save an estimated 8,126 neonatal lives compared to the 2025 baseline. This achievement would reduce the NMR to about 14 per 1,000 live births and avert an additional 5,977 child deaths and 358 maternal deaths. More than two-third (70.4%) of neonatal lives saved would be

**Data availability statement:** The data is based on LiST analysis which is publicly available and can be found from https://www.livessavedtool.org/country-data-pack.

**Funding:** The author(s) received no specific funding for this work.

**Competing interests:** The authors have declared that no competing interests exist.

attributable to four priority interventions: neonatal resuscitation (36.0%), promotion of breastfeeding practices (11.8%), case management of neonatal sepsis (11.7%) and case management of premature babies (10.9%). At this coverage level, approximately 26.9% of deaths due to prematurity, 17.9% from sepsis, and 16.1% from diarrhoea could be prevented.

## Conclusion

Nepal must prioritize highly effective interventions: neonatal resuscitation, breastfeeding, management of neonatal sepsis and case management of premature babies to accelerate progress towards the NENAP target. Our analysis indicates that achieving the specific coverage targets of 35 NENAP interventions could reduce the NMR to 14 per 1,000 live births which fall short of the NENAP targets of reducing NMR to 11 per 1,000 live births by 2035. Strengthening special newborn care unit (SNCU) in line with WHO standard is critical to improving the quality of care for small and sick newborns (SSNB) in health facilities. While strengthening newborn resuscitation and SNCU services will substantially reduce NMR, attaining the NENAP targets will ultimately require comprehensive health system reforms and effective coverage of other maternal and newborn interventions.

## Introduction

In 2022, approximately 2.3 million children died within the first 28 days of life, accounting for around 6,500 newborn deaths and representing 47% of all under-five child deaths worldwide [1]. Although the absolute number of neonatal deaths has declined over the past two decades, the proportion of neonatal mortality among under-five deaths has increased over time specially in low- and middle-income (LMIC) countries including Nepal [2]. Nepal has made significant progress in reducing child mortality in the past through a series of national strategies and programs. These include community-based interventions, facility-based interventions, demand-side financing initiatives and overarching policy frameworks like Every Newborn Action Plan (NENAP) and Nepal Safe Motherhood and Newborn Health Roadmap 2030 [3,4]. Additionally, integrated and cross-cutting interventions addressing the multiple determinants of child health have contributed crucially to these improvements. As a result, Nepal has achieved 72% reduction in under-five mortality (U5MR), a 64% decline in infant mortality (IMR), and a 58% decrease in neonatal mortality (NMR) between 1996 and 2022 [5].

Despite these notable achievements, neonatal mortality remains a critical public health problem in Nepal. According to most recent Demographic and Health Survey (DHS) survey, the NMR has stagnated at 21 deaths per 1,000 live births since 2016 [6]. Alarmingly, neonatal deaths now represent 64% of U5MR and 85% of IMR [5]. This epidemiological profile highlights a need for targeted interventions to meet Sustainable Development Goals (SDG) [7] and NENAP targets [8]. Verbal autopsy report

from recent DHS survey indicates that the primary causes of neonatal deaths in Nepal are respiratory and cardiovascular disorders of perinatal period (31%), complications related to pregnancy, labour and delivery (30%), infections specific to perinatal period (16%) and congenital malformations and deformations (7%) [5]. Further decline in U5MR and IMR are unlikely without focused measures to address these predominant causes [6]. Therefore, Nepal's health policies need to prioritize and scale up targeted interventions addressing the leading causes of neonatal deaths to effectively reduce overall child mortality and achieve the child health targets.

The NENAP serves as a country's neonatal health framework, which aims to reduce preventable newborn deaths to 11 per 1,000 live births and stillbirths to less than 13 per 1,000 total births by 2035 [8]. The plan identifies key indicators to track the progress, aligning with the global initiatives like Every Newborn Action Plan indicators and World Health Organization (WHO) recommendation. It primarily focuses on strengthening both coverage and quality of essential maternal and newborn health interventions. These include antenatal care (ANC), skilled birth attendance, immediate newborn care and resuscitation, postnatal care for mother and infants, and specialized care for small and sick newborns (SSNB), all of which will address the leading causes of neonatal mortality [9]. While the efficacy of these interventions individually is well documented, a critical evidence gap remains regarding the combined impact of simultaneously scaling up the entire package of NENAP priority interventions.

This study aims to address this critical research gap by estimating the potential impact of scaling up the combined effect of selected maternal and neonatal health interventions at the coverage targets set by NENAP for 2035. Utilizing the Lives Saved Tool (LiST), this study quantifies the number of neonatal, maternal, and child lives that could be saved through improvements in key interventions in a given time-period. The findings of this study will provide robust evidence to guide policymakers and program implementers in prioritization and scaling of high-impact maternal and neonatal health interventions, thereby informing them regarding evidence-based interventions in newborn survival policies and programs.

## Methods

We used Lives Saved Tools (LiST), —a deterministic mathematical modelling tool for priority healthcare interventions. It has been widely used to estimate the potential impact of scaling priority healthcare interventions on maternal, neonatal, and child mortality [10]. This tool provides a robust framework for understanding how scaling selected interventions could affect health outcomes [11]. In this study, LiST was used to estimate the potential impact of selected interventions identified in NENAP, assuming achievements of target coverage levels by 2035.

### Study design

We conducted the modelling using Spectrum software, version 6.1 [12] incorporating two key modules: demographic module (DemPro) and family planning module (FamPlan) [13]. The DemPro module was employed to integrate demographic projections of births and deaths, providing the population structure and fertility assumptions necessary for accurate mortality estimates. The FamPlan module estimated the impact of family planning services on birth rates and fertility patterns, which influence the number of births and subsequent mortality risks [13]. The LiST has been previously used in Nepal to estimate the maternal, newborn and under-five childhood deaths, as well as stillbirths demonstrating its applicability in this context [14,15].

### Data sources

According to recent 2021 census, Nepal's estimated population was approximately 29.1 million, with nearly three million children under 5 years of age (10.31%) and about 0.41 million infants under one year (1.41%) [16] . Using this census data as the base year, LiST projected population changes over time, incorporating population growth rates and demographic shifts. Baseline values for intervention coverage were derived from the most recent DHS survey [17], multiple

indicator cluster survey (MICS) [18], and other nationally representative data sources. For indicators lacking baseline values, proxy values were calculated based on the available regression models from previous published literatures [19,20]. The indicator definitions, along with their baseline and target values, and their sources are presented in S1 File.

## Modelling strategy

Health intervention targets specific risk factors that ultimately influence morbidity or mortality, either directly or indirectly. The impact of an intervention is calculated as the product of its effectiveness and its coverage level, assuming all other interventions remained constant. When coverage increases, more newborn and children gain protection from preventable health risks. Conversely, the decline in coverage can lead to increased deaths, while maintaining constant yields no additional lives saved [13]. This relationship underscores the critical importance of expanding and sustaining high coverage of effective interventions within health systems.

The formula to calculate the impact is:

$$Impact = (Change\ in\ intervention\ coverage) \times (Effectiveness\ of\ intervention) \times (Affected\ fraction)$$

Effectiveness is defined as the as the proportional reduction in cause-specific mortality achievable at full coverage. It was sourced from global meta-analyses and systematic reviews to ensure standardized estimates [13]. For example, zinc treatment for diarrhea has the effectiveness of 23%, which means it can prevent 23% of diarrhoeal deaths among children aged 1–59 months when implemented at 100% coverage, assuming other factors remain constant. The model assumes a proportional relationship: 50% coverage level would therefore achieve 50% of the maximum potential impact (i.e., an 11.5% reduction in diarrheal deaths in this example) [10].

We selected 35 interventions specified in the NENAP (Table 1) for inclusion in our model as these are the core monitoring indicators prioritized by the national newborn health strategy to achieve its 2035 mortality reduction targets [8]. Baseline coverage for these interventions was set at 2025 estimates, or the nearest available data, with gradual scale-up projected to reach NENAP target coverage by 2035. Table 1 below shows the baseline and target values used for this modelling study.

## Ethics issues and consent to participate

This study used publicly available secondary data and did not involve human subjects. Consequently, ethical approval and participant consent were not required, in accordance with standard research ethics guidelines for secondary data analysis.

## Results

Scaling up the coverage of 35 interventions, as presented in Table 1, could result in saving approximately 8,126 neonatal lives. This could alsoreduce the neonatal mortality rate up to 14 per 1,000 live births by 2035. The details of these projections are presented in S2 File. Neonatal resuscitation for asphyxia is projected to have greatest impact, potentially saving 2,925 lives (36%) followed by promotion of breastfeeding practices (11.8%), case management of neonatal sepsis (11.7%) and case management of premature babies (10.9%). Collectively, these four interventions could prevent more than two-third (70.4%) of neonatal deaths. This finding suggests that focusing on these four areas could yield the greatest return on investment. Fig 1 below shows the number of neonatal lives saved by different interventions.

Fig 2 presents the distribution of additional neonatal lives saved by different causes between 2025 and 2035. The finding shows that birth asphyxia accounts for the largest share (29.8%) of additional neonatal lives saved, followed by prematurity (26.9%) and neonatal sepsis (17.9%). Together these three causes represent approximately three-quarter of the additional neonatal lives saved. This finding aligns with leading causes of neonatal deaths in Nepal.

Table 2 estimates the impact of increasing the coverage of the selected interventions to the target level by 2035. It shows that achieving the targets could save the lives of 8,126 neonates, 5,977 children aged 1–59 months, and 358

**Table 1. Baseline and target coverage of selected intervention.**

| SN | Indicator | Baseline (2025) | Target (2035) |
|---|---|---|---|
| 1 | Folic acid supplementation/fortification | 4.01 | 50 |
| 2 | Safe abortion services | 48 | 60 |
| 3 | Post-abortion case management | 59.55 | 60 |
| 4 | Antenatal care | 80.2 | 95 |
| 5 | TT – tetanus toxoid vaccination | 58.2 | 95 |
| 6 | Syphilis detection and treatment | 3.4 | 95 |
| 7 | Iron Folate supplementation | 86.50 | 95 |
| 8 | Hypertensive disorders case management | 63.80 | 80 |
| 9 | Diabetes case management | 26.40 | 80 |
| 10 | MgSO4- management of pre-eclampsia | 43.30 | 80 |
| 11 | Skilled birth attendance (SBA) | 80.1 | 95 |
| 12 | Institutional delivery | 79.4 | 95 |
| 13 | Clean birth practices | 79.4 | 95 |
| 14 | Immediate assessment and stimulation | 79.4 | 95 |
| 15 | Labor and delivery management | 79.4 | 95 |
| 16 | Antibiotics for preterm premature rupture of membranes (PPRoM) | 70.33 | 95 |
| 17 | MgSO4 – management of eclampsia | 70.33 | 95 |
| 18 | Active Management of the Third Stage of Labor (AMTSL) | 70.33 | 95 |
| 19 | Induction of labor for pregnancies lasting 41+weeks | 63.26 | 95 |
| 20 | Case management of premature babies | 29.9 | 95 |
| 21 | Case management of severe neonatal infection | 71 | 100 |
| 22 | Neonatal resuscitation for asphyxia | 29.6 | 85 |
| 23 | Exclusive breastfeeding | 56.4 | 90 |
| 24 | Clean Postnatal Practices | 69.7 | 95 |
| 25 | Chlorhexidine | 51.2 | 95 |
| 26 | Complementary feeding – education only | 48.7 | 95 |
| 27 | Complementary feeding – education and supplementation | 48.7 | 95 |
| 28 | Improved water source | 94.9 | 95 |
| 29 | Water connection in the home | 36 | 50 |
| 30 | Improved sanitation | 93.8 | 95 |
| 31 | Hand washing with soap | 80.7 | 95 |
| 32 | Hygienic disposal of children's stools | 70.3 | 95 |
| 33 | Oral rehydration solution (ORS) | 38 | 90 |
| 34 | Zinc – treatment of diarrhoea | 17.9 | 90 |
| 35 | Oral antibiotics for pneumonia | 75 | 90 |

mothers. These results show that investing in newborn health has a multiplying effect. The same efforts that save new-borns also significantly reduce maternal and child deaths. This leads to a greater overall public health impact. The additional lives saved by each year is presented in the table below whose details is presented in S2 File.

## Discussion

Despite significant progress in reducing child mortality in Nepal over the past three decades, the NMR has stagnated at 21 deaths per 1,000 live births since 2016. The stagnation has caused neonatal deaths to constitute increasing proportion of infant and under-five mortality. This underscores a critical barrier to further improving in child survival. To address this, our

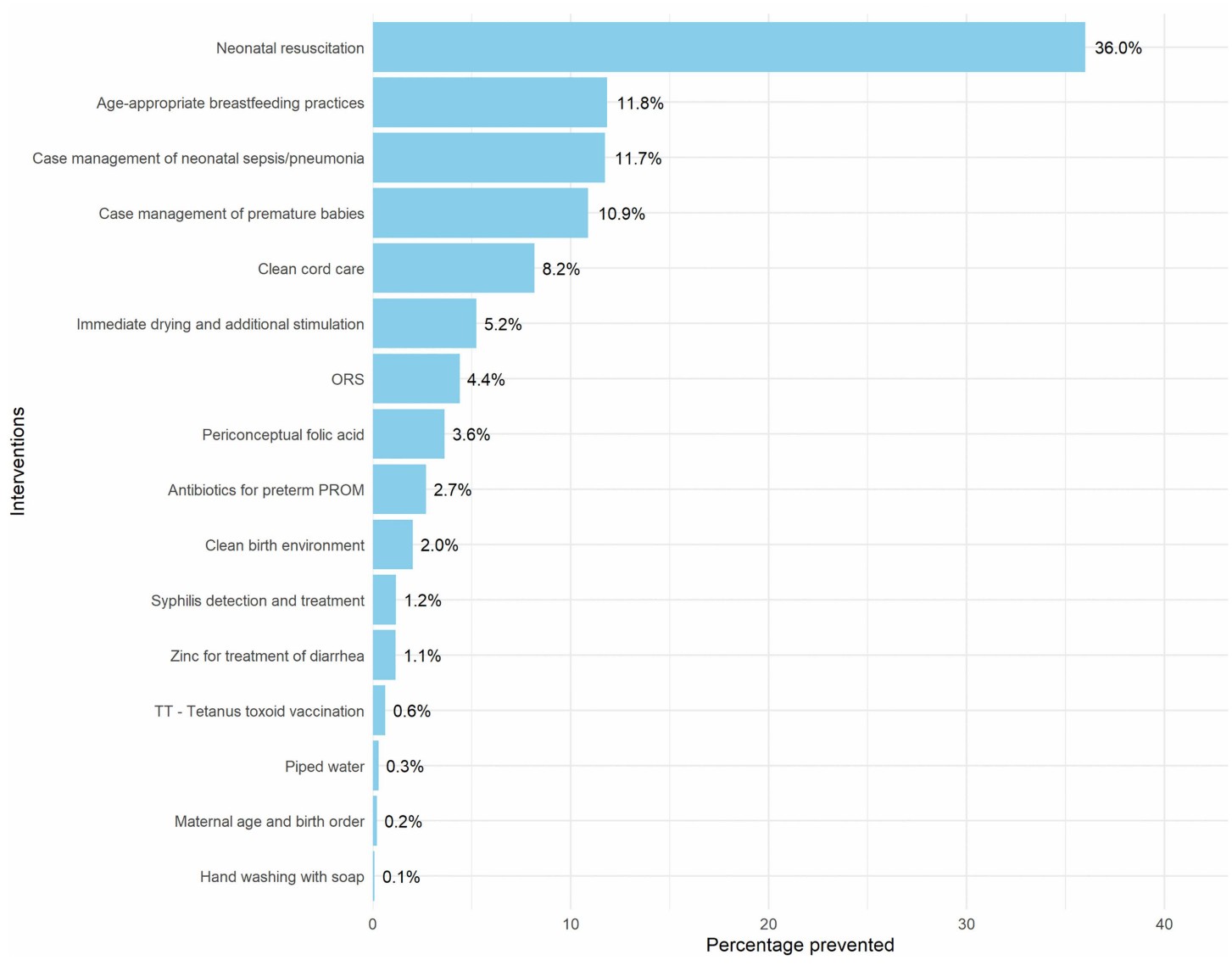

**Fig 1. Additional neonatal deaths prevented by interventions, 2025-2035 (N= 8,126).**

study used the LiST, to model the impact of scaling up 35 interventions from NENAP. Our findings show that achieving the NENAP coverage targets by 2035 could avert 8,126 neonatal deaths, offering a significant reduction in neonatal mortality. This discussion explores the implication of these findings, their alignment with global and national health priorities, and considers the opportunities and challenges for effective implementation in Nepal.

This study identified neonatal resuscitation, breastfeeding practices, case management of neonatal sepsis and case management of premature babies emerged as the most impactful interventions, collectively accounting for the potential prevention of approximately 70% of neonatal deaths. However, even with the achievements of coverage targets outlined in NENAP, the combined impact of scaling these 35 interventions is projected to be insufficient for meeting the NENAP targets of reducing the NMR to 11 per 1,000 live births by 2035. Instead, the NMR is expected to decline only upto 14 per

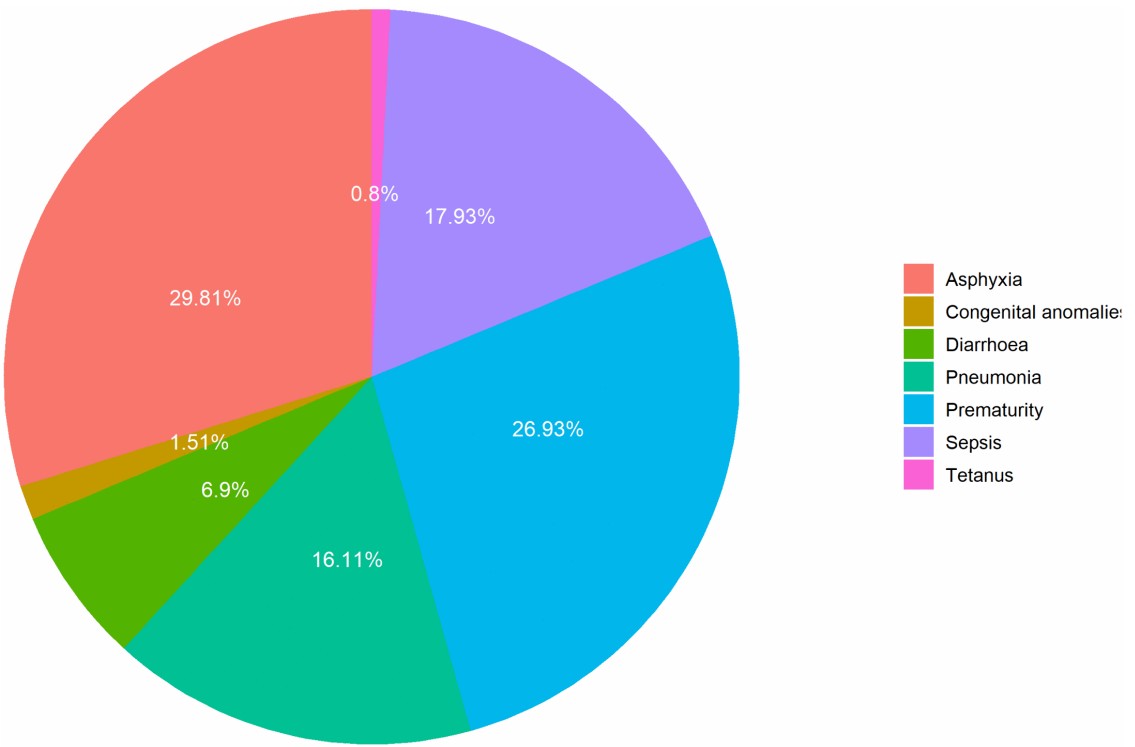

**Fig 2. Distribution of additional neonatal lives saved by causes, 2025-2035 (N= 8,126).**

**Table 2. Total number of lives saved by all interventions between 2025 and 2035.**

| SN | Age-group | Number of lives saved |
|---|---|---|
| 1 | < 1 month | 8,126 |
| 2 | 1-59 months | 5,977 |
| 3 | Maternal | 358 |

1,000 live births by 2035. These findings underscore the necessity for additional strategies beyond the current intervention package to achieve national goals.

Beyond reducing neonatal mortality, this model estimates additional benefit including the potential of saving the lives of nearly 5,977 children aged 1–59 months and 358 mothers by 2035. This ripple effect stem from improvements in antenatal care, institutional delivery, postnatal care, nutrition and WASH interventions. Similar multi-level benefits have been documented from Bangladesh and other settings where integrated maternal and neonatal health interventions have enhanced outcomes for both mothers and newborns [21,22]. However, scaling up interventions to meet NENAP 2035 targets in Nepal face several barriers such as inadequate infrastructure and equipment, shortage of trained health workers, geographical challenges, and socio-economic disparities among women [23,24]. Moreover, ensuring the quality of services is also equally critical; delivering health services without a guaranteed minimal quality level is ineffective, wasteful and unethical [25]. Therefore, quality improvement measures such as healthcare provider training and operationalizing SSNB units are essential. Strengthening facilities to deliver quality healthcare, including community-based KMC and other

proven adapted to Nepal's socio-cultural context, could provide an effective model for improving maternal and child health outcomes.

## Nepal's initiative to improve neonatal health

Nepal has made significant progress in reducing child mortality, successfully achieving Millennium Development Goal (MDG) 4 by reducing U5MR from 141 to 36 per 1,000 live births between 1990–2015 [26,27]. This achievement was partly due to community-based interventions tailored to Nepal's unique health system of Nepal. Neonatal health policies evolved from the broader child health focus of the Community-Based Integrated Management of Childhood Illness (CB-IMCI) to the more newborn-specific Community-Based Newborn Care Program (CB-NCP). These were later integrated into Community-Based Integrated Management of Neonatal and Childhood Illness (CB-IMNCI), which streamlined overlapping components to create a single, comprehensive program for managing illnesses in newborns and children under five. The substantial contribution of these community-based interventions to improving child health outcomes during the MDG period is well recognized [28]. A summary of the key policies and programs implemented to improve neonatal and child health after the year 2000 is provided in Table 3.

To meet the SDG targets of reducing NMR to 12 per 1,000 live births by 2030, Nepal must reduce its current rate by 42% [7,17]. Furthermore, the NENAP sets an even more ambitious targets of NMR of 11 per 1,000 live births, requiring a 47% reduction by 2035. Our results, which projects an NMR of 14 per 1,000 live births, provides evidence and underscore the urgent need for innovative strategies to improve neonatal survival. NENAP emphasizes high-impact intervention like KMC to manage preterm and low-birthweight (LBW) infants, which is a critical contributor to neonatal deaths in Nepal [8,38]. However, progress in implementing neonatal health infrastructure has been slow. For the past three fiscal years, the expansion of SNCUs and NICUs has stalled, highlighting a failure to turn policy into practice [23,35,39–41]. This underscores the pressing need for improved execution of neonatal health initiatives to accelerate reductions in neonatal mortality.

Although Nepal's neonatal mortality reduction program align with WHO standards for quality newborn care [42], significant gaps in implementation limit their effectiveness. For example, despite the proven benefits of KMC, its utilization remains underutilized due to inadequate integration at community-level and limited hospital capacity. The services face inadequate infrastructure, equipment, trained service providers and financial resources [43,44]. Additionally, while the guidelines exist to improve the quality of SSNB care in hospitals, overall quality of these services remains suboptimal. Moreover, the focus on inpatient care through NICU and SNCU addresses only a fraction of neonates in Nepal, leaving significant proportion of newborn underserved. This limited reach reduces the potential impact of these programs on the wider newborn population.

## Alignment of current policy interventions with LiST results

Our LiST model analysis builds on existing research by providing quantitative evidence to guide prioritization of interventions. It shows which interventions will save the most lives, offering a clear strategy for where to allocate resources. Nepal has been implementing several interventions that align closely with the findings of the LiST model. The community-focused strategy like FCHV program supports the expansion of KMC through strengthening community outreach for management of low birth weight. Capacity-building initiatives like FB-IMNCI and Point of Care Quality Improvement (PoCQI) training aims to enhance the skills of healthcare workers in managing sick newborn and reducing mortality [8,23,30]. Furthermore, the government has invested in physical infrastructure, including the establishment of NICU and SNCU, along with strengthening supply-chain system to ensure that sick newborn can access hospital care. However, the expansion of health facilities has been stagnant since fiscal year 2020/21 as shown in Fig 3.

The NENAP emphasizes improving healthcare quality by updating the health management information system (HMIS) to monitor newborn health outcomes more effectively [23,39]. The KMC guideline prioritizes capacity building,

**Table 3. Policy development for strengthening newborn care in Nepal in last two decades with their implementation status.**

| Year | Policy/Intervention | Features | Implementation status/Issues/challenges |
|---|---|---|---|
| 2002 | National Safe Motherhood Plan (2002–2017) | Strengthens infrastructure for reproductive health service delivery | Phased out |
| 2004 | National Neonatal Health Strategy | Ensures the provision of quality promotive, preventive, and curative neonatal health services | Phased out |
| 2005 | Maternity Incentive Scheme | Provides financial support to cover transportation costs for women who give birth in public health facilities, encouraging the use of skilled birth attendants. | Under implementation. The scheme expanded in to "Aama" programme in 2009 |
| | Morang Innovative Neonatal Intervention (MINI) | A pilot neonatal care project was implemented in Morang district, based on the National Neonatal Health Strategy | Phased out, this project provided valuable insights that helped develop the CB-NCP program |
| 2006 | Safe Motherhood and Newborn Health Long-Term Plan (2006–2017) | Expands BEmONC and CEmONC services while enhancing the newborn health component. | Phased out |
| | Skilled Birth Attendant Policy | Emphasizes the critical need for a skilled birth attendant to be present during every childbirth. | Under implementation, this policy aims to increase the proportion of births attended by skilled birth attendants |
| 2007 | Community-based Newborn Care Package (CB-NCP) [3] | This prioritized newborn care interventions including behaviour change communication, promotion of institutional delivery, postnatal care, management of neonatal sepsis, care of low-birth-weight newborns, prevention and management of hypothermia, and recognition and management of birth asphyxia. | Phased out and merged with CB-IMCI program in 2015. |
| 2009 | Aama program [29] | Expands the maternity incentive program to offer free delivery care at public health facilities and selected private facilities | Under implementation, the program covers all 77 districts. It provides incentives to women and health facilities to encourage service utilization. Health facilities receive reimbursement based on unit costs for delivery services, with additional payments for providing free newborn care. |
| 2011 | Chlorhexidine (CHX) for cord care [3] | Initiated application of CHX in cord to prevent sepsis which was integrated in CB-NCP package | Under implementation, the chlorhexidine (CHX) application service is currently delivered through the CB-IMNCI program. |
| 2014 | Maternal and Perinatal Death Surveillance and Response (MPDSR), Guideline | Mandatory reporting of maternal and perinatal deaths to develop a strategic plan for reducing mortality among mothers and newborns. Operated through two models: (a) Facility-Based, which reviews both maternal and perinatal deaths, and (b) Community-Based, which focuses only on maternal deaths. | Under implementation, MPDSR system collects and analyses data on maternal and perinatal deaths to identify causes and prevent future cases. The community-based model is implemented in 54 of 77 districts, while the facility-based model is active in 128 hospitals, including public, academic, medical college, and some private facilities. |
| 2015 | Community based integrated management of neonatal and childhood illnesses-CBIMNCI [3] | CB-NCP and CB-IMCI programs were integrated to CBIMNCI because about 60% of their content overlapped, leading to the development of a single training for newborn and under-five children | Under implementation, the program covers all 77 districts of Nepal, with all health facilities providing management services for newborns and children. |

*(Continued)*

**Table 3.** (Continued)

| Year | Policy/Intervention | Features | Implementation status/Issues/challenges |
|---|---|---|---|
| 2016 | Nepal's Every Newborn Action Plan (NENAP) [8] | Aims to decrease preventable newborn deaths by establishing newborn care corners in birthing centres, SNCU in district hospitals, and NICU in tertiary-level hospitals | Under implementation, the NENAP, which is based on the global Every Newborn Action Plan is being executed to reduce newborn deaths and stillbirths by improving the quality of maternal and newborn care. |
| | Facility Based IMNCI Program [30] | To enable district hospital teams to manage cases refereed from lower-level health facilities, this program provided a referral linkage for cases referred from peripheral health facilities | Under implementation and the training is being scaled up in district level hospitals. |
| | Comprehensive Newborn Care Training (Level II) package [31,32] | Six-day training package designed to equip paediatricians, medical officers and neonatal nurse with in-depth knowledge and skills to manage newborns with moderate to high-risk health conditions. The training covers resuscitation techniques, vital signs monitoring, and specialized feeding methods in Level II health facilities | Under implementation, the training is gradually being scaled up in Level II health facilities. |
| | Free Newborn Care (FNC) Program [33] | A financing scheme that provides subsides for treatment for sick newborn, offering free services that covers everything from admission to treatment. Its goal is to ensure no newborn is deprived of healthcare services due to poverty. | Under implementation, the program now covers all 77 districts of Nepal. It was merged with the "Aama" programme in 2017, integrating free newborn care into the maternal incentive scheme to provide comprehensive support for mothers and newborns. |
| 2019 | Nepal Safe Motherhood and Newborn Health Roadmap 2030 [34] | Outlines a strategic plan to lower neonatal mortality to fewer than 12 deaths per 1,000 live births by ensuring the delivery of high-quality, equitable maternal and newborn health services. | Under implementation, this plan is currently the main guiding document for maternal and neonatal health services in Nepal. |
| 2023 | Development of Nepal's model of care for SSNBs [35] | Developed through series of expert's consultation that includes 10 components: vision, political commitment, financing, human-resources, infrastructure and design, equipment and commodity, robust data system, functional referral system, linkage of maternal and newborn care, family and community involvement a support and post-discharge follow-up systems at facility and at home. | Under implementation, the program was piloted in four Level II health facilities located in Dhading, Trishuli, Sindhuli, and Hetauda districts. |
| | NENAP Implementation plan (2023–2030) [36] | Outlines a plan to improve maternal and newborn healthcare and reduce newborn deaths and stillbirths | Under implementation |
| | Development of KMC Guideline [37] | Standardize and improve care for premature babies and infants with a low birth weight across the country | Under implementation |
| 2024 | Orientation Package for SNCU Level II Care Facilities | Aims to orient and build the capacity of newborn service providers in essential knowledge and skills for SSNB care at SNCUs. | Under implementation |
| | Development and Piloting of Infant and Family Centered Developmental Care: From Hospital to Home | This model emphasizes infant-centered care, by creating a nurturing, low-stress sensory environment and developmental needs. It promotes collaborative partnership between the family and service providers to ensure holistic care, aiming for optimal newborn well-being and development. | Under implementation, the model was piloted at Paropakar Maternity and Women's Hospital and Bharatpur Hospital. |

Abbreviations: BEmONC: Basic Emergency Obstetric and Newborn Care; CEmONC: Comprehensive Emergency Obstetric and Newborn Care; SNCU: Special newborn care units; NICU: Neonatal intensive care unit; SSNB: Small and Sick Newborn.

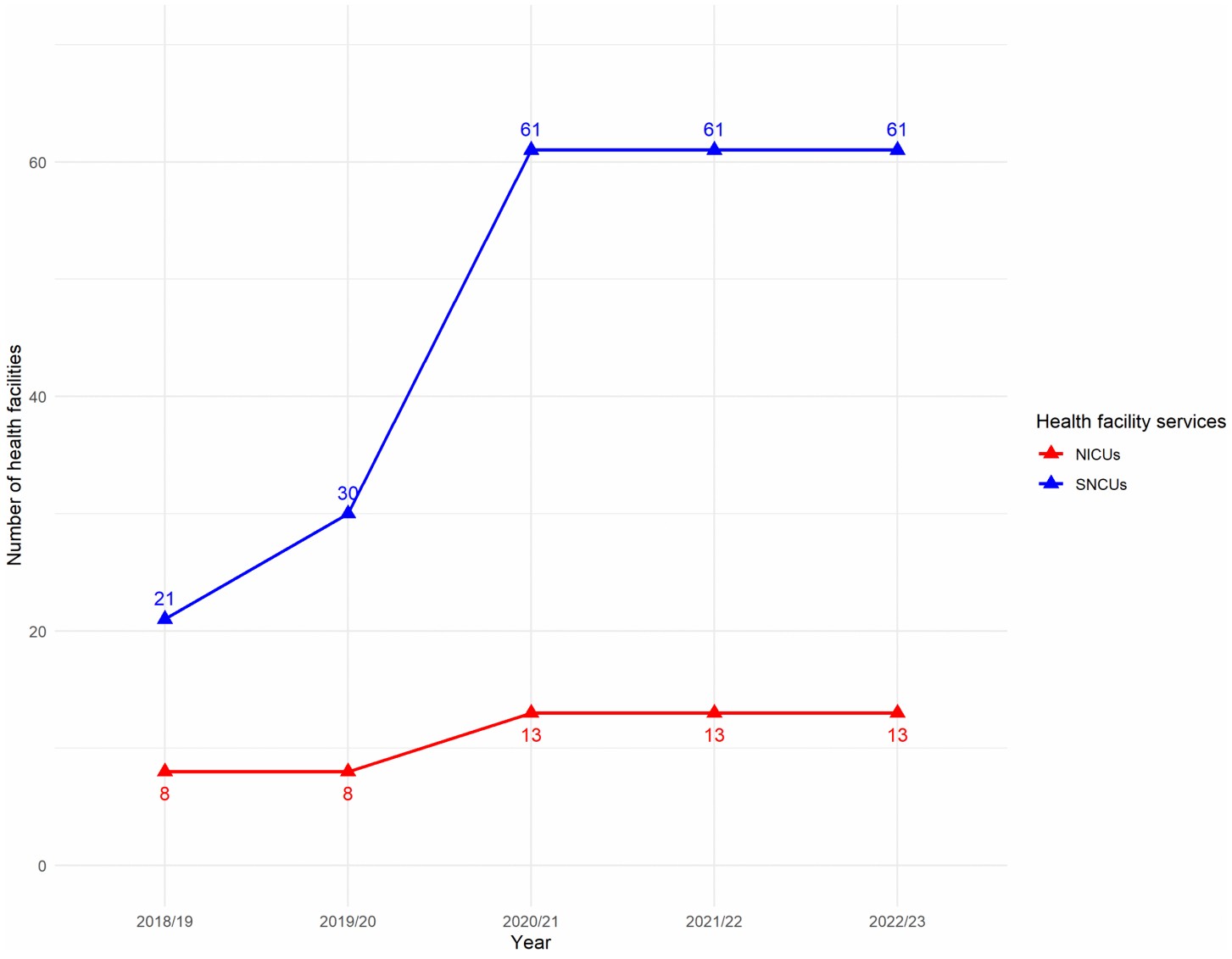

**Fig 3. Expansion of SNCU and NICU facilities over time (Fiscal year 2018/19 to 2022/23).**

infrastructure upgrades, community programs, hypothermia monitoring using BEMPU bracelets, and public awareness campaigns to increase demand for KMC services [37]. Since 2023, Family Welfare Division has allocated budgets for training across all 61 SNCU hospitals, with 378 nurses and 79 medical officers trained by 2024. Collectively, these efforts strengthen healthcare workers' capacity, strengthen health system functionality, and engage communities—key components necessary for scaling up effective newborn care in Nepal.

Despite sustained efforts to improve newborn care, several critical misalignments hinder in effective implementation of neonatal health interventions in Nepal. First, the absence of dedicated CB-IMNCI officers at provincial and municipal levels, along with insufficient specialized staff in SNCUs, NICUs and KMC units creates coordination gaps across the three tiers of governments during implementation [39]. This challenge is exacerbated by weakness in the referral system, causing delays in timely and effective treatment for newborn [8]. Second, Nepal's financing model for the FNC program,

which includes Packages A (USD 7.41), B (USD 14.81), and C (USD 37.04), to cover newborn care costs at birthing centres, SNCUS and NICUs [45]. This amount does not cover the full operational costs [45,46] or reduce out-of-pocket expenditure for families [47], thereby limiting both the access and quality of newborn care. Third, frequent supply chain interruption; including periodic stockouts of essential supplies, and inadequate engagement of private sector disrupts the uninterrupted supply and delivery of newborn care services [35,39]. For instance, outcome of neonatal resuscitation depends on uninterrupted supply of equipment and quality care, which Nepal often struggles to maintain. Fourth, Nepal lacks robust data system to track SNCU training activities conducted by federal and provincial governments; ongoing training programs have rarely been systematically evaluated for their effectiveness in provider skill retention, equipment sufficiency or readiness of health facilities. Lastly, while community-based interventions are contextually appropriate for Nepal, they lacks capacity to manage complex cases like LBW and small for gestational age (SGA) cases, which requires inpatient care [35]. These challenges — workforce shortages, financing constraints, supply chain interruptions, inadequate training evaluation, and limitations of community care pose major barriers to scaling up and sustaining effective newborn health interventions in Nepal.

## Strength and limitation

There are some strengths of this paper. This study estimates the combined impact of selected interventions when delivered at the coverage levels estimated by NENAP. Thus, this model incorporates maximum number of interventions possible in this analysis. Additionally, for those indicators whose baseline values are lacking in DHS or MICS survey, this study calculated the proxy estimates as suggested by Spectrum manual [11] and regression equations suggested by previous studies [19,20].

Despite these strengths, this paper has some limitations that should be considered when interpreting the findings. First, we used the most recent coverage data available for Nepal, with DHS 2022 [5] and MICS 2018 data [18] (for WASH indicators), both of which predate the 2025 baseline period. Second, the effect size of individual interventions in the LiST model are drawn from global or regional evidence; as a result, national projections may not fully reflect Nepal's unique context. Variation in healthcare access, service utilization, disease burden, cultural practices and health system characteristics could lead to overor underestimation of intervention impacts in our model. Third, the modelling primarily relies on DHS data, which focuses on coverage, but not account for service quality which is critical factor for health system outcome. Fourth, this study does not include sensitivity analyses or alternative coverage scenarios (example different coverage level), limiting assessment of how changes in coverage might affect outcomes.

## Conclusion

This study assessed the impact of interventions on neonatal mortality in Nepal, identifying neonatal resuscitation, breast feeding, case management of neonatal sepsis, and case management of premature babies as the most effective for saving additional neonatal lives between 2025 and 2035. Despite Nepal's remarkable progress in child health in the past, the NMR has remained stagnated at around 21 deaths per 1,000 live births since 2016 underscoring the urgent need to scale up highly effective interventions. Existing initiatives, such as NENAP, FNC programs, and the expansion of SNCUs and NICUs provide a strong foundation. However, persistent gaps in coverage, quality, and infrastructure hinders the progress. Achieving targeted coverage requires addressing several critical barriers including inadequate healthcare workforce, geographic inaccessibility, and financial constraints. Quality improvement measures like training service providers, equipping facilities, and ensuring uninterrupted supply of commodities which are crucial for achieving the success. Comprehensive health system reforms, like robust and sustainable financing, private sector engagement, and community education, are equally critical to bridge the gaps in implementation. In conclusion, while strengthening newborn resuscitation and SNCU services can significantly reduce NMR, achieving the NENAP targets will ultimately require integrating these high-impact neonatal interventions into a stronger, equitable and resilient health system.

## Supporting information

**S1 File. Summary of indicators, their definitions, baseline and targets, and data sources.**
(DOCX)

**S2 File. Additional tables for detail results by year, 2025–2035.**
(DOCX)

## Author contributions

**Conceptualization:** Geha Nath Khanal.

**Data curation:** Geha Nath Khanal, Nisha Giri.

**Formal analysis:** Geha Nath Khanal.

**Investigation:** Geha Nath Khanal, Nisha Giri, Deepak Jha, Dipak Raj Chaulagain.

**Methodology:** Geha Nath Khanal, Deepak Jha, Dipak Raj Chaulagain.

**Software:** Geha Nath Khanal.

**Supervision:** Dipak Raj Chaulagain.

**Validation:** Deepak Jha, Dipak Raj Chaulagain.

**Visualization:** Geha Nath Khanal, Nisha Giri.

**Writing – original draft:** Geha Nath Khanal.

**Writing – review & editing:** Geha Nath Khanal, Nisha Giri, Deepak Jha, Dipak Raj Chaulagain.

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
