## [Decision Letter · Decision Letter 0]

27 May 2025

PONE-D-25-15453Where to Invest in Neonatal Survival Programs for Achieving the Targets of Sustainable Development Goals in Nepal?  A Modelling Study Using Lives Saved Tool Through Scaling Key InterventionsPLOS ONE?

Dear Dr. Khanal,

Thank you for submitting your manuscript to PLOS ONE. After careful consideration, we feel that it has merit but does not fully meet PLOS ONE’s publication criteria as it currently stands. Therefore, we invite you to submit a revised version of the manuscript that addresses the points raised during the review process.

**ACADEMIC EDITOR:**

**Comments:**

Reviewer #1: The topic addressed in this paper is highly relevant for Nepal and other low- and middle-income countries (LMICs).

Specific Comments:

Page 1 – Introduction:

While the introduction provides useful background, it omits key factors contributing to Nepal’s progress in maternal and neonatal health. Specifically, the role of integrated programs involving family planning, water, sanitation, and hygiene (WASH), as well as community-based approaches is missing.

Methods Section:

• The rationale for selecting 90% coverage for all nine interventions is not clearly explained. Why were this specific threshold chosen, and why was it uniformly applied across all interventions?

• The decision to assign equal weight to all nine interventions requires justification. Not all interventions may have an equivalent impact on neonatal mortality, so the use of equal weightage should be discussed.

• The inclusion of the Family Planning (FamPlan) and AIDS Impact Module (AIM) is unclear. Why were these modules used in the context of neonatal health and how they contribute to the analysis?

• The assumption of a 0% baseline for fetal growth restriction management and Kangaroo Mother Care (KMC) is unrealistic and should be revisited or supported with evidence.

Key Issue Regarding SDG Target:

Even under the assumption of achieving 90% coverage for all interventions, the SSDG) target of reducing neonatal mortality rate (NMR) to 12 deaths per 1,000 live births by 2030 is not met. The manuscript does not adequately address why 90% coverage was chosen if it does not align with achieving the SDG target. Given that the title suggests alignment with SDG goals, this discrepancy should be discussed more thoroughly.

Discussion Section:

The discussion is overly lengthy and lacks focus, which may reduce reader engagement. Summarizing or condensing context-specific information to improve clarity and maintain relevance for an international audience will strengthen the paper.

Minor Revisions:

• Page 8, Lines 152–153: Citation is missing; please provide appropriate references.

• Page 17: Expand “FCHV” upon first use to ensure clarity for all readers.

Language and Grammar:

The manuscript would benefit from thorough language and grammar editing.

Reviewer #2: Thank you for reaching out to review this important manuscript. The study is very relevent for Nepal considering stagnation of neonatal mortality since 2016, and need to achieve ambitious targets of Sustainable Development Goal and Nepal Every Newborn Action Plan. It would be very valuable to idedentify effective, feasible and affordable interventions to reduce neonatal mortality in Nepal. Use of LiST is a very good approach as it provides roboust model for estimating potential lives saved due to various interventions with varied level of coverage.

As this study is based on a mathematical modelling, it is very important for the authors to check assumptions and apply their judgement of the applicability and limitations of the model, as well as to make attempts to improve the model, values and assumptions. Though the authors have noted in the limitation section, the biggest weakness of this study is to rely on default values for 2024. As per my understanding, the default values provided by the LiST software is only for the reference purpose and the authors are encouraged to identify reliable sources of data to best estimate the value for the baseline and to use those values for the model. For example, as also noted by the authors, 0% fetal growth restriction detection and management and 0% case management of premature babies with KMC in 2024 is not reflective of the true sitation. Another example is, 75.6% baseline coverage of neonatal rescuscitation is over estimation of the coverage considering Nepal's insittutional delivery rate, availibility of skilled birth attendants during birth, skill retention of birth attendants, availibility of newborn rescuitation kits in health facilities.

Authors noted in the Limitation section (line 358) that "LiST relies on default effectiveness estimates derived from global or regional data" but the Spectrum Manual Page 7 guides "Make any modifications necessary to the default baseline data", which reflects that change of default value in the software is possible. As a scientific work, I strongly advise the authors to use reliable estimate for the baseline value, which otherwise will provide under/over estimation of the effect and may misguide the policy-makers and program-planners. For example, this may likely to affect the three priority interventions (case management of premature babies, management of neonatal sepsis, and neonatal resuscitation) or their order.

Also, as noted by the authors, "the assumption of uniformly scaling up all interventions 90% coverage by 2030 oversimplifies implementation challenges and may not be realistic". This is also very important aspect of the study to re-consider. I would suggest considering the targets in key policy documents such as SDG, Nepal Health Sector Strategic Plan, Nepal Every Newborn Action Plan to estimate the best possible level (target) by 2030.

The "Discussion" section is lengthy and not very coherent. Please revise it with a proper logical flow. Considering Nepal's vulnerability (e.g 2015 earthquake, COVID pandemic), please also talk about vulnerablity to achieve 2030 targets and actions needed to make the newborn health system more resilient. Try to evaluate how prepared Nepal's health system is to address newborn health priorities in case of any shocks.

As addressing these comments may require substaintial changes in the manuscript, I would recommed "major revision" for this manuscript.

Few minor corrections:

Line # 69-70: I would not consider Neonatal Strategy as an intervention. Its a strategic document guideing priority interventions. As Nepal still relies on 2004 Strategy for Neonatal health (though there are newer documents guiding newborn health such as Safe Motherhood and Newborn Health Roadmap and Nepal Every Newborn Action Plan). So, please differentiate between policy documents and actual interventions. Please provide an overview of existing key policy documents (with key dates of endorsement) guiding newborn health in Nepal.

Line # 91: rephrase the word "advocates" as it may not fit well with WHO's mandate/role

Sentences after line# 296 is repetation of sentences from line # 267, which might be a copy-paste error. Please check and update.

We look forward to receiving your revised manuscript.

Kind regards,

Sabita Tuladhar

Academic Editor

PLOS ONE

Journal Requirements:

1. Please ensure that your manuscript meets PLOS ONE's style requirements, including those for file naming. The PLOS ONE style templates can be found at https://journals.plos.org/plosone/s/file?id=wjVg/PLOSOne_formatting_sample_main_body.pdf and https://journals.plos.org/plosone/s/file?id=ba62/PLOSOne_formatting_sample_title_authors_affiliations.pdf .

Reviewers' comments:

Reviewer's Responses to Questions

**Comments to the Author**

1. Is the manuscript technically sound, and do the data support the conclusions?

Reviewer #1: Yes

Reviewer #2: Partly

2. Has the statistical analysis been performed appropriately and rigorously?

Reviewer #1: I Don't Know

Reviewer #2: No

3. Have the authors made all data underlying the findings in their manuscript fully available?

Reviewer #1: Yes

Reviewer #2: Yes

4. Is the manuscript presented in an intelligible fashion and written in standard English?

Reviewer #1: Yes

Reviewer #2: Yes

Reviewer #1: The topic addressed in this paper is highly relevant for Nepal and other low- and middle-income countries (LMICs).

Specific Comments:

Page 1 – Introduction:

While the introduction provides useful background, it omits key factors contributing to Nepal’s progress in maternal and neonatal health. Specifically, the role of integrated programs involving family planning, water, sanitation, and hygiene (WASH), as well as community-based approaches is missing.

Methods Section:

• The rationale for selecting 90% coverage for all nine interventions is not clearly explained. Why were this specific threshold chosen, and why was it uniformly applied across all interventions?

• The decision to assign equal weight to all nine interventions requires justification. Not all interventions may have an equivalent impact on neonatal mortality, so the use of equal weightage should be discussed.

• The inclusion of the Family Planning (FamPlan) and AIDS Impact Module (AIM) is unclear. Why were these modules used in the context of neonatal health and how they contribute to the analysis?

• The assumption of a 0% baseline for fetal growth restriction management and Kangaroo Mother Care (KMC) is unrealistic and should be revisited or supported with evidence.

Key Issue Regarding SDG Target:

Even under the assumption of achieving 90% coverage for all interventions, the SSDG) target of reducing neonatal mortality rate (NMR) to 12 deaths per 1,000 live births by 2030 is not met. The manuscript does not adequately address why 90% coverage was chosen if it does not align with achieving the SDG target. Given that the title suggests alignment with SDG goals, this discrepancy should be discussed more thoroughly.

Discussion Section:

The discussion is overly lengthy and lacks focus, which may reduce reader engagement. Summarizing or condensing context-specific information to improve clarity and maintain relevance for an international audience will strengthen the paper.

Minor Revisions:

• Page 8, Lines 152–153: Citation is missing; please provide appropriate references.

• Page 17: Expand “FCHV” upon first use to ensure clarity for all readers.

Language and Grammar:

The manuscript would benefit from thorough language and grammar editing.

Reviewer #2: Thank you for reaching out to review this important manuscript. The study is very relevent for Nepal considering stagnation of neonatal mortality since 2016, and need to achieve ambitious targets of Sustainable Development Goal and Nepal Every Newborn Action Plan. It would be very valuable to idedentify effective, feasible and affordable interventions to reduce neonatal mortality in Nepal. Use of LiST is a very good approach as it provides roboust model for estimating potential lives saved due to various interventions with varied level of coverage.

As this study is based on a mathematical modelling, it is very important for the authors to check assumptions and apply their judgement of the applicability and limitations of the model, as well as to make attempts to improve the model, values and assumptions. Though the authors have noted in the limitation section, the biggest weakness of this study is to rely on default values for 2024. As per my understanding, the default values provided by the LiST software is only for the reference purpose and the authors are encouraged to identify reliable sources of data to best estimate the value for the baseline and to use those values for the model. For example, as also noted by the authors, 0% fetal growth restriction detection and management and 0% case management of premature babies with KMC in 2024 is not reflective of the true sitation. Another example is, 75.6% baseline coverage of neonatal rescuscitation is over estimation of the coverage considering Nepal's insittutional delivery rate, availibility of skilled birth attendants during birth, skill retention of birth attendants, availibility of newborn rescuitation kits in health facilities.

Authors noted in the Limitation section (line 358) that "LiST relies on default effectiveness estimates derived from global or regional data" but the Spectrum Manual Page 7 guides "Make any modifications necessary to the default baseline data", which reflects that change of default value in the software is possible. As a scientific work, I strongly advise the authors to use reliable estimate for the baseline value, which otherwise will provide under/over estimation of the effect and may misguide the policy-makers and program-planners. For example, this may likely to affect the three priority interventions (case management of premature babies, management of neonatal sepsis, and neonatal resuscitation) or their order.

Also, as noted by the authors, "the assumption of uniformly scaling up all interventions 90% coverage by 2030 oversimplifies implementation challenges and may not be realistic". This is also very important aspect of the study to re-consider. I would suggest considering the targets in key policy documents such as SDG, Nepal Health Sector Strategic Plan, Nepal Every Newborn Action Plan to estimate the best possible level (target) by 2030.

The "Discussion" section is lengthy and not very coherent. Please revise it with a proper logical flow. Considering Nepal's vulnerability (e.g 2015 earthquake, COVID pandemic), please also talk about vulnerablity to achieve 2030 targets and actions needed to make the newborn health system more resilient. Try to evaluate how prepared Nepal's health system is to address newborn health priorities in case of any shocks.

As addressing these comments may require substaintial changes in the manuscript, I would recommed "major revision" for this manuscript.

Few minor corrections:

Line # 69-70: I would not consider Neonatal Strategy as an intervention. Its a strategic document guideing priority interventions. As Nepal still relies on 2004 Strategy for Neonatal health (though there are newer documents guiding newborn health such as Safe Motherhood and Newborn Health Roadmap and Nepal Every Newborn Action Plan). So, please differentiate between policy documents and actual interventions. Please provide an overview of existing key policy documents (with key dates of endorsement) guiding newborn health in Nepal.

Line # 91: rephrase the word "advocates" as it may not fit well with WHO's mandate/role

Sentences after line# 296 is repetation of sentences from line # 267, which might be a copy-paste error. Please check and update.

**Do you want your identity to be public for this peer review?** For information about this choice, including consent withdrawal, please see our Privacy Policy

Reviewer #1: **Yes: ** Sabita Tuladhar

Reviewer #2: **Yes: ** Deepak Paudel

---

## [Author Response · Author response to Decision Letter 1]

14 Sep 2025

Dear Editor and Reviewer,

We sincerely appreciate the time and effort you have dedicated to reviewing our manuscript. We are grateful for your insightful comments and constructive suggestions, which have significantly improved the quality of our work.

In response to the reviewers’ feedback, we have carefully addressed each point raised. Please find the attached reviewer comments file.

---

## [Decision Letter · Decision Letter 1]

19 Oct 2025

PONE-D-25-15453R1Where to Invest in Neonatal Survival Programs in Nepal?  A Modelling Study Using Lives Saved Tool Through Scaling Key InterventionsPLOS ONE?

Dear Dr. Khanal,

Thank you for submitting your manuscript to PLOS ONE. After careful consideration, we feel that it has merit but does not fully meet PLOS ONE’s publication criteria as it currently stands. Therefore, we invite you to submit a revised version of the manuscript that addresses the points raised during the review process.

We look forward to receiving your revised manuscript.

Kind regards,

Sabita Tuladhar, PhD, MHealSc, MA

Academic Editor

PLOS ONE

Journal Requirements:

Additional Editor Comments (if provided): None.

Reviewers' comments:

Reviewer's Responses to Questions

**Comments to the Author**

Reviewer #1: All comments have been addressed

Reviewer #3: (No Response)

2. Is the manuscript technically sound, and do the data support the conclusions?

Reviewer #1: Yes

Reviewer #3: Yes

3. Has the statistical analysis been performed appropriately and rigorously?

Reviewer #1: Yes

Reviewer #3: Yes

4. Have the authors made all data underlying the findings in their manuscript fully available?

Reviewer #1: Yes

Reviewer #3: Yes

5. Is the manuscript presented in an intelligible fashion and written in standard English?

Reviewer #1: Yes

Reviewer #3: Yes

Reviewer #1: This is an important and timely paper that can contribute to reducing neonatal deaths in Nepal and monitoring progress toward the SDGs. Thank you for addressing all comments carefully. A thorough review of the language, in-text citations, and references would further strengthen the manuscript.

Reviewer #3: Overall Summary

This paper provides a clear, data-driven analysis of neonatal mortality in Nepal using the LiST model to estimate the impact of scaling up national priority interventions. It is strong in its policy relevance, methodological transparency, and integration of national and global health frameworks. The analysis effectively identifies high-impact interventions and links them to actionable policy strategies. There are a few targeted areas for improvement, such as tightening the writing, polishing grammar and some phrasing, and strengthening the connection between results and policy implications, that would enhance coherence and impact.

Abstract

The abstract effectively addresses a critical public health issue - neonatal mortality in Nepal - within a clear global and national policy context. It presents well-organized sections with quantitative, model-based estimates that make the projected impact tangible and actionable. The identification of key priority interventions adds strong practical relevance and supports evidence-based decision-making. There are a few minors areas of improvement you could consider:

• Minor grammatical and phrasing issues need correction for clarity.

• Methods section could briefly clarify use of proxy estimates and modeling assumptions.

• Integrating a few points of how findings can inform policy or program implementation would strengthen conclusion.

• It would be helpful to address the gap between projected NMR (14/1,000) and the NENAP target (11/1,000).

Introduction

The introduction provides a clear, data-driven rationale for the study, effectively situating neonatal mortality within both the global and national (Nepalese) context. It demonstrates strong use of recent data and policy references, highlighting Nepal’s past progress, remaining challenges, and alignment with national and international frameworks such as NENAP, SDGs, and WHO recommendations. The logical flow from problem statement to research gap and study aim is coherent and well-grounded in evidence. A few minor items that would strengthen this section:

• Correct minor grammatical and phrasing errors (e.g., line 111: “….effect of selected maternal and neonatal health interventions to the coverage targets set by NENAP for 2035” would be better phrased as “….effect of selected maternal and neonatal health interventions at the coverage targets set by NENAP for 2035”; e.g., line 117 “setting the evidence-based interventions in newborn survival policies and programs” is a bit awkward – perhaps use “informing” or “guiding” the integration of evidence-based interventions.

• Consider streamlining a few sections to improve readability and maintain focus on key points (e.g., summarize or group Nepal’s MNH programs in lines 73-79 instead of a full list; lines 85-98 could be streamlined a bit as it repeats similar ideas about the need for targeted interventions).

• Strengthen the transition to the study aim by clarifying the research gap.

• Briefly note limitations of past interventions to justify the study’s relevance.

Methods

The methods section clearly describes the modeling tool (LiST) and software used, providing sufficient detail about the data sources and modules involved. It explains the modeling strategy and assumptions well, including how intervention effectiveness and coverage translate into impact. The inclusion of baseline and target coverage data offers transparency and supports reproducibility. A few minor areas to improve clarity and readability:

• Some sentences are long and complex; breaking them into shorter, clearer statements would improve readability (e.g., lines 121–124).

• Minor grammatical issues (e.g., line 125 “affect in health outcomes” to “affect health outcomes”).

• The explanation of effectiveness and impact calculations could be streamlined to avoid redundancy (lines 159–170).

• A short add on to clarify the rationale for selecting the 35 interventions—why these and not others? – would be helpful. For example, “We selected…….model, prioritizing those identified as key drivers of neonatal survival and aligned with national targets for 2035“ or perhaps “…focusing on those with established effectiveness and significant potential impact on neonatal, maternal, and child mortality.”

Results

The results section clearly quantifies the projected impact of scaling up interventions, providing specific numbers of lives saved and highlighting the most impactful interventions. It effectively uses percentages to contextualize the relative contributions of key interventions and causes of death. There is a comprehensive presentation of data. A few minor items to strengthen the results presentation:

• Correct the broken reference ("Error! Reference source not found") to ensure all citations and figures are properly linked and accessible (line 198).

• Avoid repetition in listing the four key interventions twice in consecutive sentences (lines 192–197). For example, the second reference (starting at line 194) could combine into one clear sentence: “These four interventions together could prevent more than two-thirds (70.4%) of neonatal deaths.”

• A few points of awkward phrasing: e.g., line 202 “The finding shows that birth asphyxia accounts the largest share (29.8%)” could be “Birth asphyxia accounts for the largest share (29.8%) of additional neonatal lives saved.”

• To enhance reader understanding, include brief interpretations or implications of the results, rather than just presenting numbers.

Discussion

The discussion is comprehensive and demonstrates a strong command of both the national policy landscape and the study’s implications. It effectively links the model results to real-world challenges in Nepal’s neonatal health system, referencing existing programs, infrastructure, and health system constraints. The integration of evidence and policy history adds depth, and the section maintains alignment with the study’s objectives and global standards. A few items could be addressed to strengthen the discussion:

• Consider places to streamline the discussion section and address minor grammatical issues. For example, can condense overlapping explanations (e.g., repeated statements about Nepal’s progress and NMR stagnation in lines 215–220 and 261–266) or simplify long sentences for readability.

• Focus the policy table (Table 3) by summarizing key milestones or grouping related initiatives rather than listing every program in detail. For instance, combine overlapping entries like CB-NCP, CB-IMCI, and CB-IMNCI.

• You could strengthen the analytical discussion by more clearly distinguishing between what the LiST model adds and what prior research or policy already shows.

Conclusion

The conclusion effectively summarizes key findings and policy implications, maintaining consistency with the results and discussion. It ends with a clear call for a comprehensive, system-wide response, which is a strong way to close the paper. You could consider ending with a lightly stronger emphasis, such as: “Ultimately, achieving the NENAP targets will depend on integrating high-impact neonatal interventions within a strengthened, equitable health system.”

**Do you want your identity to be public for this peer review?** For information about this choice, including consent withdrawal, please see our Privacy Policy

Reviewer #1: **Yes: ** Sabita Tuladhar

Reviewer #3: No

---

## [Author Response · Author response to Decision Letter 2]

30 Oct 2025

REVIEWER 1

This is an important and timely paper that can contribute to reducing neonatal deaths in Nepal and monitoring progress toward the SDGs. Thank you for addressing all comments carefully. A thorough review of the language, in-text citations, and references would further strengthen the manuscript.

Response: We would like to thank you for your assessment and review of previous version of our manuscript. We are pleased that the previous revisions that we had made after your valuable review and feedback were satisfactory.

We have tried our best in reviewing the language and references as suggested and have performed a through proofreading of the manuscript to correct grammatical errors. We have checked and added some references (increased from 43 to 46 now) which were missing in the previous version of manuscript. Furthermore, we have also added supplementary table 4 which was missing in the earlier version.

REVIEWER 3

This paper provides a clear, data-driven analysis of neonatal mortality in Nepal using the LiST model to estimate the impact of scaling up national priority interventions. It is strong in its policy relevance, methodological transparency, and integration of national and global health frameworks. The analysis effectively identifies high-impact interventions and links them to actionable policy strategies. There are a few targeted areas for improvement, such as tightening the writing, polishing grammar and some phrasing, and strengthening the connection between results and policy implications, that would enhance coherence and impact.

Response: Thank you for your evaluation and feedback. We have tried our best to improve in grammar, sentence structure and coherence between results and policy implications as suggested.

Abstract

The abstract effectively addresses a critical public health issue - neonatal mortality in Nepal - within a clear global and national policy context. It presents well-organized sections with quantitative, model-based estimates that make the projected impact tangible and actionable. The identification of key priority interventions adds strong practical relevance and supports evidence-based decision-making.

Response: Thank you for your observation and evaluation.

There are a few minors areas of improvement you could consider:

1. Minor grammatical and phrasing issues need correction for clarity.

Response: Thank you for suggestion. We have revised it accordingly.

2. Methods section could briefly clarify use of proxy estimates and modelling assumptions.

Response: Thank you for your suggestions. We have CORRECTED accordingly.

3. Integrating a few points of how findings can inform policy or program implementation would strengthen conclusion.

Response: Thank you for suggesting this important point which was missing in the previous version of manuscript. We have CORRECTED accordingly.

4. It would be helpful to address the gap between projected NMR (14/1,000) and the NENAP target (11/1,000).

Response: Thank you for your critical observation. We have CORRECTED accordingly.

Introduction

The introduction provides a clear, data-driven rationale for the study, effectively situating neonatal mortality within both the global and national (Nepalese) context. It demonstrates strong use of recent data and policy references, highlighting Nepal’s past progress, remaining challenges, and alignment with national and international frameworks such as NENAP, SDGs, and WHO recommendations. The logical flow from problem statement to research gap and study aim is coherent and well-grounded in evidence. A few minor items that would strengthen this section:

Response: Thank you for your observation.

5. Correct minor grammatical and phrasing errors (e.g., line 111: “….effect of selected maternal and neonatal health interventions to the coverage targets set by NENAP for 2035” would be better phrased as “….effect of selected maternal and neonatal health interventions at the coverage targets set by NENAP for 2035”; e.g., line 117 “setting the evidence-based interventions in newborn survival policies and programs” is a bit awkward – perhaps use “informing” or “guiding” the integration of evidence-based interventions.

Response: Thank you for your observation. We have CORRECTED accordingly.

6. Consider streamlining a few sections to improve readability and maintain focus on key points (e.g., summarize or group Nepal’s MNH programs in lines 73-79 instead of a full list; lines 85-98 could be streamlined a bit as it repeats similar ideas about the need for targeted interventions).

Response: Thank you for your feedback. We have CORRECTED accordingly. (Page 4 )

7. Strengthen the transition to the study aim by clarifying the research gap.

Response: Thank you for your critical observation. We have CORRECTED accordingly. The expanded section reads as following: (Page 4, Line 106 to 111)

“While the efficacy of these interventions individually is well documented, a critical evidence gap remains regarding the combined impact of simultaneously scaling up the entire package of NENAP priority interventions. Our study aims to provide policymakers with the evidence by quantifying how many lives could be saved and to what extent such a scale-up could close the gap between current trends and national targets.”

8. Briefly note limitations of past interventions to justify the study’s relevance.

Response: Thank you for your feedback. We have CORRECTED accordingly.

Methods

The methods section clearly describes the modeling tool (LiST) and software used, providing sufficient detail about the data sources and modules involved. It explains the modeling strategy and assumptions well, including how intervention effectiveness and coverage translate into impact. The inclusion of baseline and target coverage data offers transparency and supports reproducibility. A few minor areas to improve clarity and readability:

Response: Thank you for your observation.

9. Some sentences are long and complex; breaking them into shorter, clearer statements would improve readability (e.g., lines 121–124).

Response: Thank you for your feedback. We have CORRECTED accordingly.

10. Minor grammatical issues (e.g., line 125 “affect in health outcomes” to “affect health outcomes”).

Response: Thank you for your feedback. We have CORRECTED accordingly.

11. The explanation of effectiveness and impact calculations could be streamlined to avoid redundancy (lines 159–170).

Response: Thank you for your feedback. We have revised accordingly. The revised section reads as follows: (Page 8, Line 161 to 172).

The formula to calculate the impact is:

Impact = (Change in intervention coverage) X (Effectiveness of intervention) X (Affected fraction)

Effectiveness is defined as the as the proportional reduction in cause-specific mortality achievable at full coverage, was sourced from global meta-analyses and systematic reviews to ensure standardized estimates [13]. For example, zinc treatment for diarrhea has the effectiveness of 23%, which means it can prevent 23% of diarrhoeal deaths among children aged 1-59 months when implemented at 100% coverage, assuming other factors remain constant. The model assumes a proportional relationship: 50% coverage level would therefore achieve 50% of the maximum potential impact (i.e., an 11.5% reduction in diarrheal deaths in this example) [10].

12. A short add on to clarify the rationale for selecting the 35 interventions—why these and not others? – would be helpful. For example, “We selected…….model, prioritizing those identified as key drivers of neonatal survival and aligned with national targets for 2035“ or perhaps “…focusing on those with established effectiveness and significant potential impact on neonatal, maternal, and child mortality.”

Response: Thank you for your feedback. We have revised accordingly. The revised section reads as follows: (Page 8, Line 174 to 176).

We selected 35 interventions specified in the NENAP (Table 1) for inclusion in our model as these are the core monitoring indicators prioritized by the national newborn health strategy to achieve its 2035 mortality reduction targets [8].

Results

The results section clearly quantifies the projected impact of scaling up interventions, providing specific numbers of lives saved and highlighting the most impactful interventions. It effectively uses percentages to contextualize the relative contributions of key interventions and causes of death. There is a comprehensive presentation of data. A few minor items to strengthen the results presentation:

Response: Thank you for your observation.

13. Correct the broken reference ("Error! Reference source not found") to ensure all citations and figures are properly linked and accessible (line 198).

Response: Thank you for your critical observation. We have CORRECTED it accordingly.

14. Avoid repetition in listing the four key interventions twice in consecutive sentences (lines 192–197). For example, the second reference (starting at line 194) could combine into one clear sentence: “These four interventions together could prevent more than two-thirds (70.4%) of neonatal deaths.”

Response: Thank you for your observation. This has been CORRECTED in the revised manuscript.

15. A few points of awkward phrasing: e.g., line 202 “The finding shows that birth asphyxia accounts the largest share (29.8%)” could be “Birth asphyxia accounts for the largest share (29.8%) of additional neonatal lives saved.”

Response: Thank you for your important observation. We have CORRECTED it accordingly.

16. To enhance reader understanding, include brief interpretations or implications of the results, rather than just presenting numbers.

Response: Thank you for your important observation. We have revised it accordingly. (Page 10-11 (Result section)).

Discussion

The discussion is comprehensive and demonstrates a strong command of both the national policy landscape and the study’s implications. It effectively links the model results to real-world challenges in Nepal’s neonatal health system, referencing existing programs, infrastructure, and health system constraints. The integration of evidence and policy history adds depth, and the section maintains alignment with the study’s objectives and global standards.

Response: Thank you for your evaluation.

A few items could be addressed to strengthen the discussion:

17. Consider places to streamline the discussion section and address minor grammatical issues. For example, can condense overlapping explanations (e.g., repeated statements about Nepal’s progress and NMR stagnation in lines 215–220 and 261–266) or simplify long sentences for readability.

Response: Thank you for your critical observation. We have revised it accordingly.

18. Focus the policy table (Table 3) by summarizing key milestones or grouping related initiatives rather than listing every program in detail. For instance, combine overlapping entries like CB-NCP, CB-IMCI, and CB-IMNCI.

Response: We sincerely appreciate this critical suggestion to streamline Table 3. We agree that focusing the table is crucial for clarity.

In our initial design, we considered grouping related initiatives. However, the discussion in our team, we made a deliberate choice to list key programs like CB-IMCI, CB-IMNCI, and CB-NCP separately to accurately represent the chronological evolution and specific focus of Nepal's neonatal health policy over the past two decades. Presenting them individually allows readers to see how the national strategy has shifted and refined its approach, moving from a broader child health focus (CB-IMCI) to a more newborn-specific one (CB-NCP) and integrating them to CBIMNCI later. We believe this detailed view is valuable for understanding the historical context of policy priorities. In the revised manuscript we have explained the historical development of community-based interventions and their evolution in paragraphs. The expanded section reads as following: (Page 13, Line 261-269)

“……This achievement was partly due to community-based interventions tailored to Nepal’s unique health system of Nepal. neonatal health policies evolved from the broader child health focus of the Community-Based Integrated Management of Childhood Illness (CB-IMCI) to the more newborn-specific Community-Based Newborn Care Program (CB-NCP). These were later integrated into Community-Based Integrated Management of Neonatal and Childhood Illness (CB-IMNCI), which streamlined overlapping components to create a single, comprehensive program for managing illnesses in newborns and children under five. …..”

19. You could strengthen the analytical discussion by more clearly distinguishing between what the LiST model adds and what prior research or policy already shows.

Response: Thank you for your critical feedback. We have revised it accordingly.

Conclusion

The conclusion effectively summarizes key findings and policy implications, maintaining consistency with the results and discussion. It ends with a clear call for a comprehensive, system-wide response, which is a strong way to close the paper. You could consider ending with a lightly stronger emphasis, such as: “Ultimately, achieving the NENAP targets will depend on integrating high-impact neonatal interventions within a strengthened, equitable health system.”

Response: Thank you for your valuable feedback. We have CORRECTED as suggested

---

## [Editor Report · Decision Letter 2]

6 Nov 2025

Where to Invest in Neonatal Survival Programs in Nepal? A Modelling Study Using Lives Saved Tool Through Scaling Key Interventions

PONE-D-25-15453R2

Dear Dr. Khanal,

We’re pleased to inform you that your manuscript has been judged scientifically suitable for publication and will be formally accepted for publication once it meets all outstanding technical requirements.

Kind regards,

Sabita Tuladhar, PhD, MHealSc, MA

Academic Editor

PLOS ONE
---

## [Editor Report · Acceptance letter]

PONE-D-25-15453R2

PLOS ONE

Dear Dr. Khanal,

I'm pleased to inform you that your manuscript has been deemed suitable for publication in PLOS ONE. Congratulations! Your manuscript is now being handed over to our production team.

Kind regards,

on behalf of

Dr. Sabita Tuladhar

Academic Editor

PLOS ONE